# Studies on the Wooden Box Containing the "Marco Polo" Bible

**Francesco Augelli** 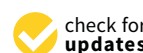

Dipartimento di Architettura e Studi Urbani, Politecnico di Milano, Observatory for Conservation of Wood Works, P.zza L. da Vinci 26, 20133 Milano, Italy; francesco.augelli@polimi.it; Tel.: +39-02-2399-5807

**Abstract:** The aim of this paper is to present the results of research undertaken on a wooden box that holds an important historical book: a hand Bible handwritten in the thirteenth century. Tradition connects this Bible to the name of Marco Polo (Venice, 1254–1324), who was supposedly the owner—the book possibly accompanied him on his travels (1262 and 1271) to China. The Bible is of fine workmanship and written on thin parchment, and its container—along with a yellow silk cloth—are preserved in the ancient and prestigious Laurentian Library in Florence. The manuscript was in very poor condition and was being restored during the period of study (2011). Surveys were carried out to determine the place and period of manufacture of the box, and to determine if it was contemporary to or later than the manuscript it contained or whether it was made in China or Europe. An additional aim of the work was to demonstrate that a fast and inexpensive in situ survey under imperfect time and space conditions was possible using in-depth observation and simple tools as well as a portable microscope, all performed without sampling. During the restoration process, a team of experts used instruments helpful in determining the chemical composition of the paper and related ink. Other specialists studied the paleography of the text. The results indicate that the Bible is definitely from the same period as Marco Polo. Nothing excludes the possibility that Marco Polo may have seen it or lived not too far from this manuscript, which traveled in a small wooden box, wrapped in a precious yellow silk cloth.

**Keywords:** wooden object; cultural heritage; history; direct sources; macroscopic analysis

## 1. Introduction

*Short Historical Information about the So-Called "Marco Polo" Bible*

The so-called "Marco Polo" Bible (Figure 1) is a pocket-sized 16.5 cm × 11 cm manuscript produced in southern France on veal parchment before the mid-thirteenth century and entrusted to one of the Franciscan missions between 1244 and the beginning of the fourteenth century. They reached China to ask the Khan of the Mongols and the imperial courts for an alliance, the relationships of which we know the details of through exchanges of letters and reports [1].

According to some studies [1], the "Marco Polo" Bible was owned by an important family of Ch'ang-shu. It was found and then purchased by (or received as a gift from) a Belgian Jesuit missionary, Philippe Couplet (Mechelen, 1623–Goa, 1693). The family had owned the Bible for more than four hundred years. The circumstances surrounding the discovery of the manuscript, its condition at that time, and its transportation to Italy remain obscure to this day.

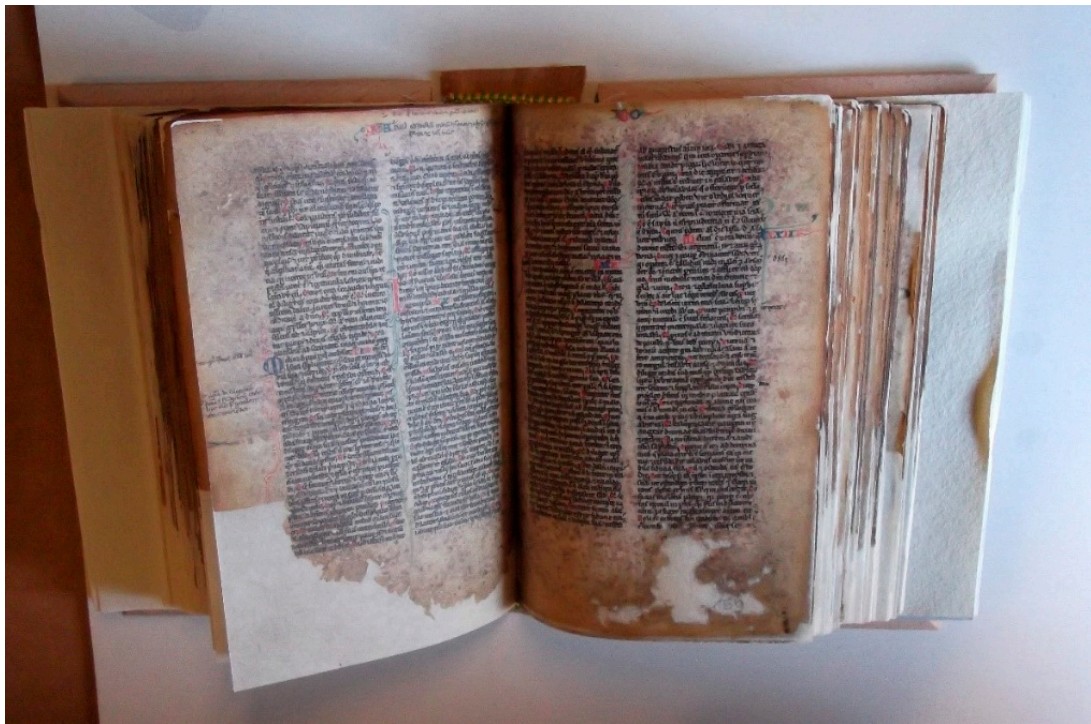

**Figure 1.** The Bible during restoration.

During Couplet's journey to Italy between 1685 and 1686, which included Rome and Florence, he donated about fourteen old books to the Duke Cosimo III de' Medici who later transferred them to the Laurentian Medici Library in Florence [1].

Among these volumes was included the precious Bible. Late seventeenth century documents in the archive, which seem to refer to the Bible, never mention the wooden box or anything about the manuscript's condition.

During the entire Yuan dynasty (1271–1368), and later, after the conquest of Beijing by Zhu Yuanzhang in 1368, during the Ming dynasty (1368–1644), we do not know where this Bible was kept or anything about its movements—the definition given by Couplet of "the Bible of Marco Polo" makes no mention of possible bearers, but it does specify a long presence of the book in China [2].

We also do not know who wrapped the sacred book in the yellow silk that still protects it—a cloth tinted with a color forbidden outside the Chinese court, which could possibly suggest that the keeper of this first and extraordinary witness of the contact between the cultures of Latin Europe and China was of high rank. Alternatively, it could suggest that someone in the Florentine library, due to the bad condition of the cover and of the book's many pages, thought to wrap the Bible with a precious silk handkerchief around the beginning of the eighteenth century [2].

In 1957, the Bible drew the attention of the medievalist Boleslaw Szczesniak from the University of Notre Dame, Indiana, USA, who wrote an article about what he considered a "Franciscan Bible of the fourteenth century". A Byzantinist from the University of Athens took note and, in an exhibition organized in 2008, the Bible was subjected to new studies. In 2011, a restoration was undertaken by the Foundation for the religious sciences of Bologna and the Institute of the Italian Encyclopaedia [2].

The parchment from which the Bible was made was photographed using VIS and UV light and underwent FTIR and Raman spectroscopic analysis. In addition, microphotographs were taken using stereo microscopy and scanning electron microscopy. Proteomics analysis in mass spectrometry of a fragment already detached from the manuscript was also conducted. "Proteomics" is the systematic analysis of all protein sequences and the ways in which proteins interact with each other and in

tissues; this investigation allowed us to precisely identify the nature of the parchment. For this reason, some small fragments were sent to the laboratory [3].

Paleographic studies were able to confirm the manufacturing period of the Bible but not his "owner", which in any case seems unlikely to have been Marco Polo. The numerous glosses (notes) present in the text's margins are more an indication that it was part of an ecclesiastic's belongings. We cannot exclude the possibility that the person was a clergyman who accompanied Marco Polo [4].

Unfortunately, the indirect sources were not useful for determining ownership.

## 2. Materials and Methods

*Research Indicators and Study Phases*

To define the box's age and place of manufacture, reference has been made to possible indirect sources (historical data), and especially the following four primary sources:

- Historical information;
- Technological and construction characteristics;
- Wood species;
- Stylistic elements.

In describing the construction characteristics, we also wanted to consider information regarding the state of conservation.

The study's conclusions were deduced from these indicator elements, using the circumstantial type method.

Unfortunately, as mentioned above, the indirect sources (i.e., historical written documents) do not mention the box. For this reason, the only way to reply to requests about the age of the box and its site of production was to use a direct source. That task involved the direct in-depth reading of the artifact. The signs on the item allowed us to translate them into useful information—a methodology frequently also used by architects for historical buildings and by art critics for dating works and attributing the designer or artist.

We did not want to touch the important box. This approach was also adopted for ethical reasons, which require the conservator-restorer to "respect the aesthetic, historic and spiritual significance and the physical integrity of the cultural heritage entrusted to her/his care"[1].

The task involved working hurriedly and in an inadequate space (in a little room inside the Florence library) with the sole purpose of determining if the box was made in China during the same period as the Bible.

## 3. Results

*3.1. Technological and Construction Characteristics*

The box measures[2] 135.5 mm × 186 mm and has a height of 68.8 mm. The four sides are joined with four "dovetail" joints on each corner. The wall thickness is about 22 mm. The bottom, in a single 4.4-mm-thick piece, is glued flush to the lower edge, while the cover (7.2 mm thickness) is made to slide along the guides formed in the upper inner sides. The left outer side and the cover have ink inscriptions [5].

---

[1]  E.C.C.O. European Confederation of Conservator-Restorers' Organizations, Code of Ethics, adopted at the General Assembly, Brussels, 7 March 2003. Citation from Article 5 of Obligations towards Cultural Heritage.
[2]  All measurements were obtained directly using a plastic centesimal gauge to avoid surfaces' damages.

On the left side, sentences about the library location of the box containing the Bible are written:

"*Trovasi in questa cassetta (da riporre nel Pluteo III)*

*1. una Bibbia Latina (membr., sec. XIII), [molto? mutila?] impo(?)rata,*

*2. un docum.to firmato Philippus Couplet S.J.*

*3. altro documento di Andrea Giulianelli*

*4. copia di quest'ultimo fatta da C.R.*[3]

*5. un verbale di ricognizione firmato, G.Vacco il 13 Giugno 1923*" (Figure 2)[4]

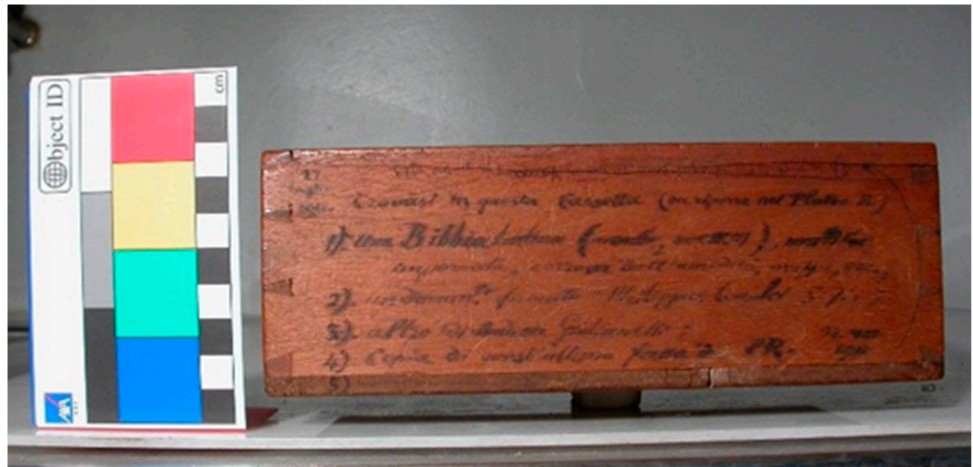

**Figure 2.** The outer side of the left side of the box.

On top left, the date: *27 Luglio 1912*.
On the lid, at the top left corner, the letter *J*.
In central position, the words:

"*C.I.*
*Appartenente al*
*Pluteo III*
*Biscioni*
*Catal. p. 121.*
*BIBLIA*
*APUD ETHNICUM*
*IN CHINA INVENTA.*" [5] (Figure 3)

---

[3]  "*It is in this box (to put in the 3rd sit):*

*1. a Latin Bible (parchment, XIII century) [very? mutilates?]*
*2. a document signed Philippus Couplet S.J.*
*3. another document of Andrea Giulianelli*
*4. copy of the previous document mad by C.R.*
*5. a reconnaissance report signed by G. Vacco 13 June 1923*

[4]  This note is written in the opposite direction and is not well readable in visible light. For this we used UV light and so the words became better readable.
[5]  Belonging to the 3rd sit Biscioni, Catalogue p.121. PAGAN BIBLE FOUND IN CHINA.

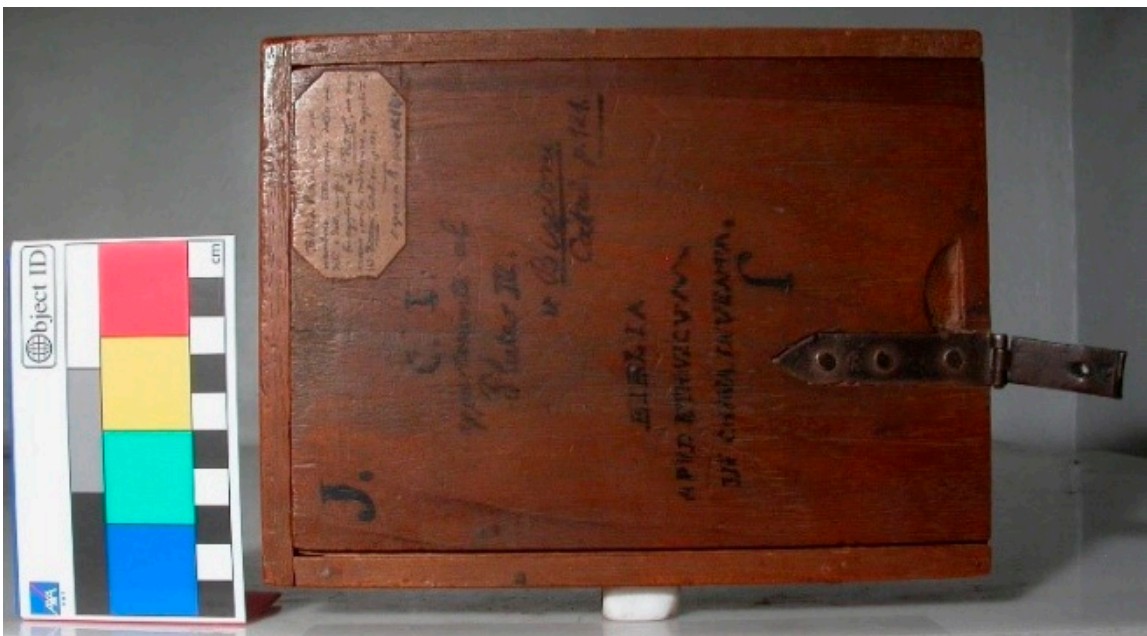

**Figure 3.** The lid's outer side.

The inscription is followed by an inverted *J* letter.

The information written on the left side of the box documented that in 1923 the Latin Bible, already dated as XIII century, was in very bad condition.

The lid is made up of three asymmetrical boards glued on the edges and has a glued paper label on its upper outer right side (Figure 4). Furthermore, the ends towards the front side have a recessed lanceolate hinge, fastened with three nails along the axis, and a plate of 1.4 mm thickness hinged for a key closure of the lid[6].

On the front side of the lid, a small niche has been made in a semicircle for the insertion of fingers in order to facilitate the sliding along the sides of the box.

On the outer side of the bottom, one rectangular carving of 2 mm is present for the insertion of a dovetail metal plate and the three holes along the axis.

It is possible that this is the track of the restraint system of the box to the bench of Pluteo III (in the Laurentian consultation big hall), as also indicated by the glued undated cartouche on the lid, where the following is written:

*"Bibbia latina (sec. XIII:*
*membran.: tutta corrosa dall'umi-*
*dità e dalla muffa).*
*Fu aggiunta al* <u>*Plut. III°*</u>*, ma senza*
*numero, con la radicazione 'Capsula I'.*
*Ved.* <u>*Biscioni*</u>*, Catal. ecc. p.121*
*Si apre con la chiave N.14."* [7] (Figure 4)

---

6   The key is no longer available.
7   "Latin Bible (XIII century, parchment, all rotted by humidity and mold). Was added to 3rd sit but without number, with the reference 'Capsula I'. Widow Biscioni, Catalogue ecc.ecc. p.121.It could be open with the key N.14." This label is probably of the same year as the sentence written on the left side.

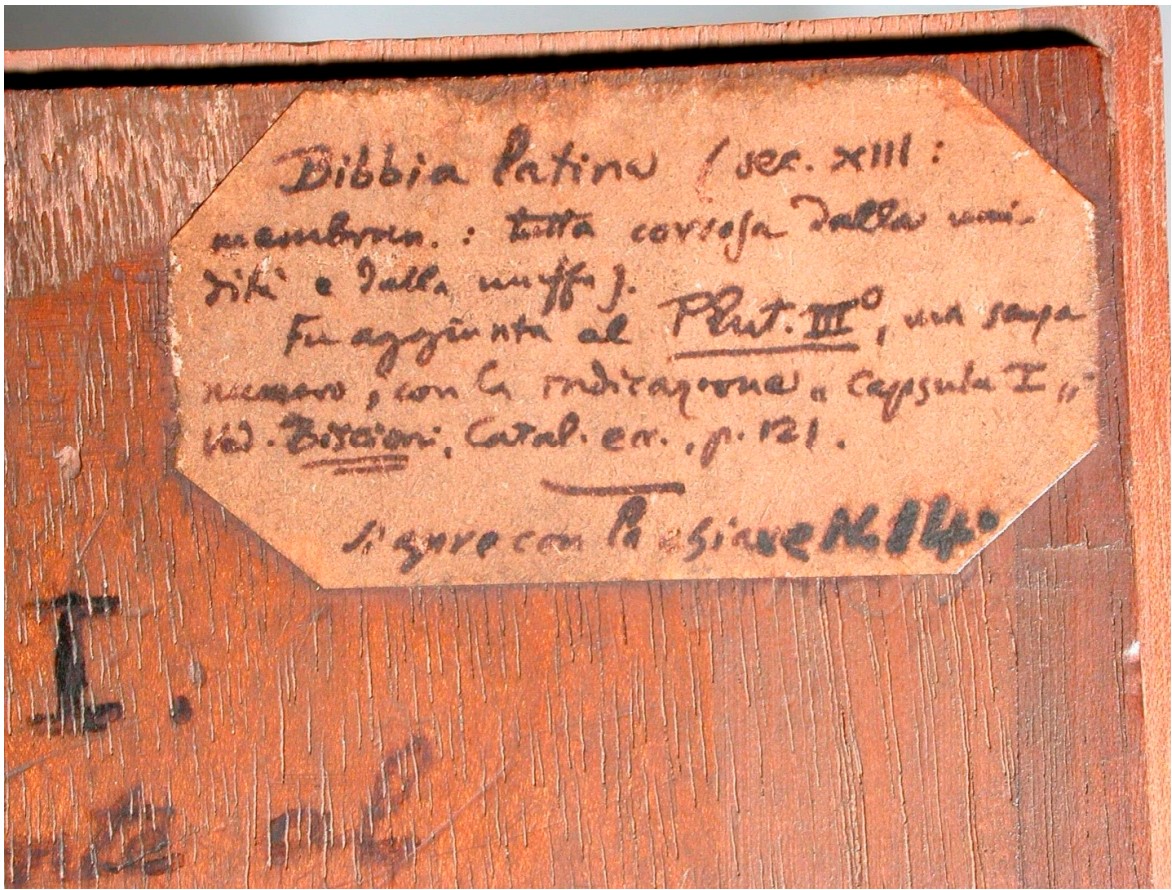

**Figure 4.** Detail of the label glued on top of the lid.

The label on the lid refers to the different locations of the Bible inside the Library and again about its poor conditions, "deteriorated by humidity and mold".

On the frontal face, a lock with an iron cartouche of 1.4 mm thickness and two ink inscriptions (*C.* and *I.*) are present. (Figure 5)

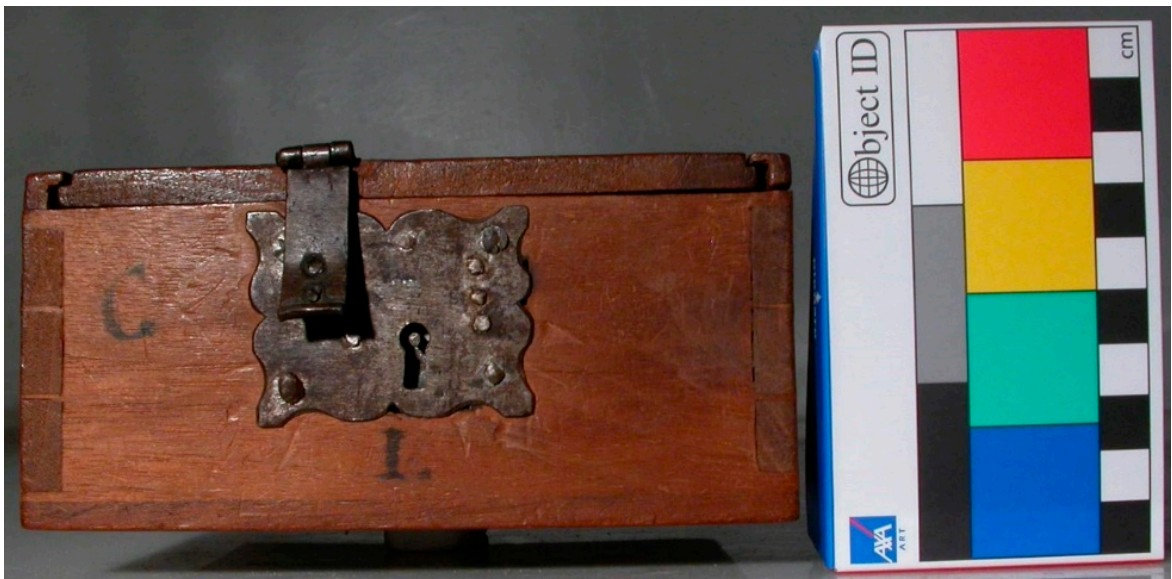

**Figure 5.** Front view of the box.

After the survey of the general characteristics and writings on the box, the survey proceeded to define—again without samplings—the nature of the glue, of the varnish, and of the species of wood. The used glue returns a notable white light under Wood light conditions. We observed that all connections and joints are bonded with organic glue (i.e., ox glue). Moreover, emissions under UV light of all the outside surfaces of the box suggested that they are coated with a thin, transparent, glossy finish that is amber in color, and probably shellac-based [6] (Figures 6–8).

The inner sides do not present any visible finishing coating.

The conservation condition of the box is good. Negligible previous attacks by wood-boring anobiid insects can be observed (judging from the characteristic of three exit holes visible on the surface, probably *Oligomerus ptilinoides*) [7].

Fissures due to shrinkage characterize the bottom of the box and the left side (Figure 8).

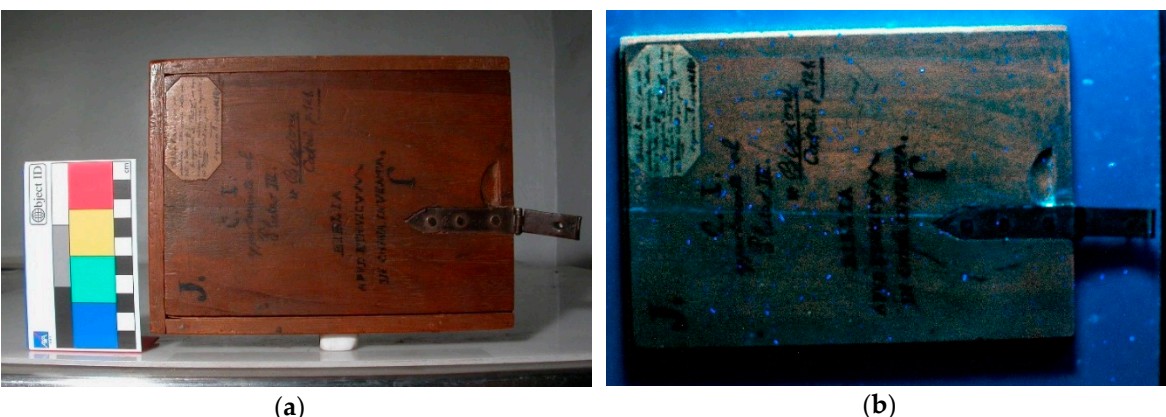

| (**a**) | (**b**) |

**Figure 6.** (**a**) The lid's outer side under visible light; (**b**) under Wood's light.

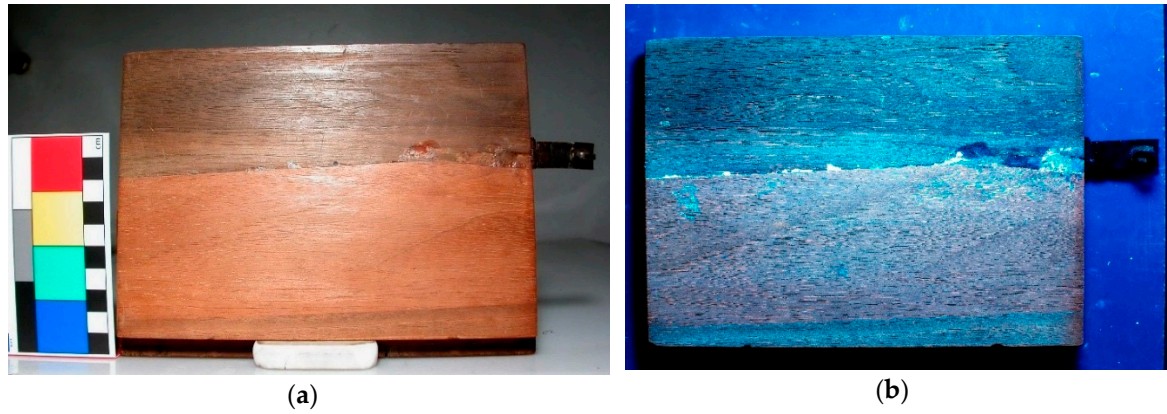

**Figure 7.** (**a**) The lid's inner side under visible light; (**b**) under Wood's light.

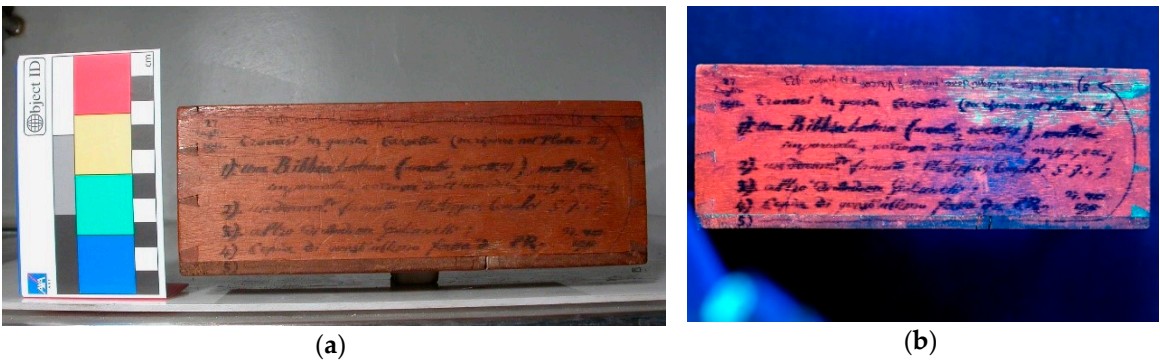

**Figure 8.** (**a**) The outer side of the left side under visible light; (**b**) under Wood's light.

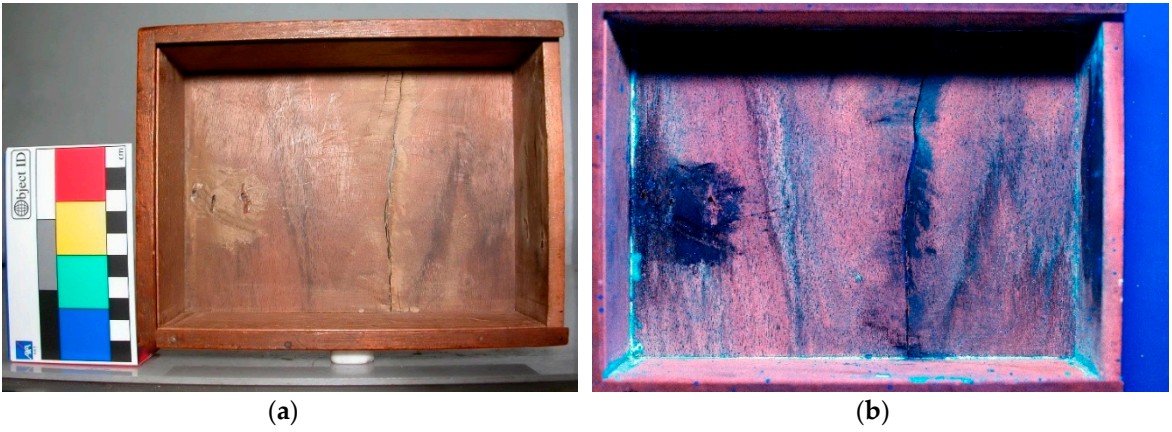

**Figure 9.** (**a**) Inside view under visible light; (**b**) under Wood's light.

The inner side of the bottom and of the front are also characterized by the presence of clumsy groutings along the fissures, possibly performed to close the holes of the plate with a "dovetail" metal plate. By observation with Wood's light, these groutings may be characterized—as it was customary in the Western countries until the beginning of the XX century—by a mix of Boulogne gypsum and rabbit glue (Figures 9 and 10).

On the bottom, a wide fissure that runs orthogonally and with an irregular pattern from side to side also highlights the repair attempts by the insertion of nails along the edge, which favored the formation of more fissures (Figure 11).

The fissure on the top edge of the left side of the box, which originates from the top joint and runs up to the opposite side, has been repaired by using animal glue whose identification is conceivable from the intense fluorescence emitted under ultraviolet light exposition.

No technological or construction element argues in favor of an attribution of the box to a Chinese art craft, nor to the period of the medieval Bible.

Unfortunately, despite the research carried out, no technological or diagnostic studies have been found on similar wooden boxes that could have been useful as a reference for this study.

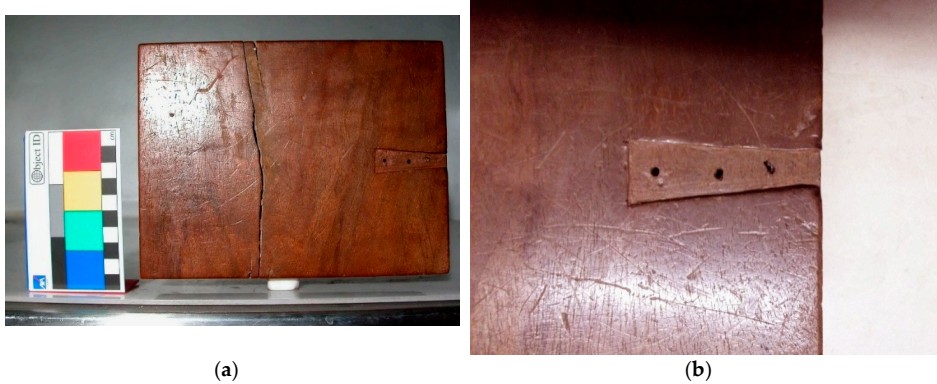

(**a**)                (**b**)

**Figure 10.** (**a**) General view of the outer side of the bottom; (**b**) detail of the outer side of the bottom with the rectangular carving for the insertion of a dovetail metal plate.

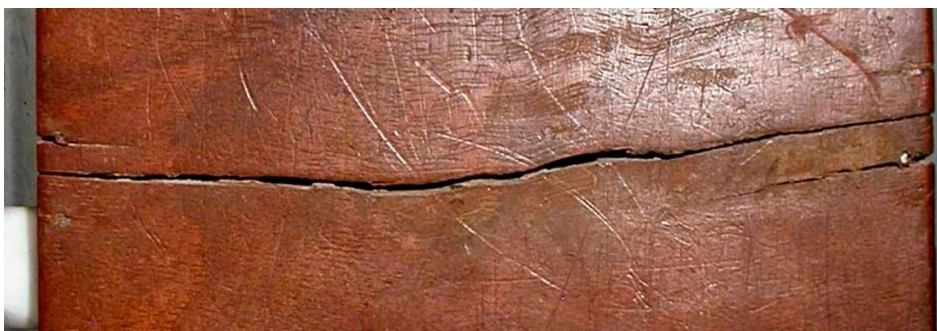

**Figure 11.** Detail of the outer side of the bottom. Irregular fissures—some of them favored by the insertions of "modern" nails.

### 3.2. Wood Species

Due to the historical importance of the item, we did not proceed with a withdrawal of material for an analysis of wood species through thin-section microscopy.

To obtain the species of wood, at first glance attributable to European or exotic species, all surfaces of all the sides of the box were observed—particularly the corners where the cross sections are well-visible (Figures 12 and 13) [7].

The species determination was then conducted in accordance with the standard UNI 11118, using only the macroscopic method [8]. This standard describes the criteria and the limits for the identification of wooden species on artifacts of historical, artistic, and archaeological interest. In point 3.2 of this standard, it is established about the macroscopic identification that is an identification based on the detection of wood characters which are macroscopically visible, or with magnification

not greater than 10×, such as: color, veining, texture, and any anatomical characters. In point 4, the standard states that the identification must be performed in successive phases, starting from the macroscopic examination. Furthermore, if there is a lack of enough anatomical characters to guarantee the macroscopic identification or if the identification is not satisfactory for specific needs, it is possible to proceed with a microscopic identification. If this cannot be done directly on the artifact, its execution must be subject to the assessment of the admissibility of the levy of a sample [8].

The observations done with a portable digital microscope[8] used with a direct light showed—on both the longitudinal (Figure 14) and transversal surfaces (Figure 15)—the presence of porous wood with whitish sapwood that clearly stands out from the heartwood brown, sometimes streaked with darker veins.

The growth rings could be identified quite easily. The vessels, which are macroscopically visible, have a slightly variable diameter, slightly and gradually decreasing moving from the early-wood area to the late-wood area [7,9].

On the radial surface, they appear to be of elliptical-roundish shape, isolated or gathered together in groups of two or three, arranged in radial rows and sometimes containing tylosis [7,9].

On the longitudinal surface simple perforations of the transverse walls of the conducting tissues can be noticed [7,9].

Macroscopic and microscopic characteristics led to the identification of wood belonging to the Juglandaceae family and attributable to European Walnut (*Juglans regia*) that is widely distributed in Europe [9]. It is important to underline that the identified wood species can indicate the provenance of the employed wood, but no certainties on the production place could be deduced. In any case, the *Juglans regia* wood mostly indicates a European production of the box, as it was frequently used to produce precious objects' containers.

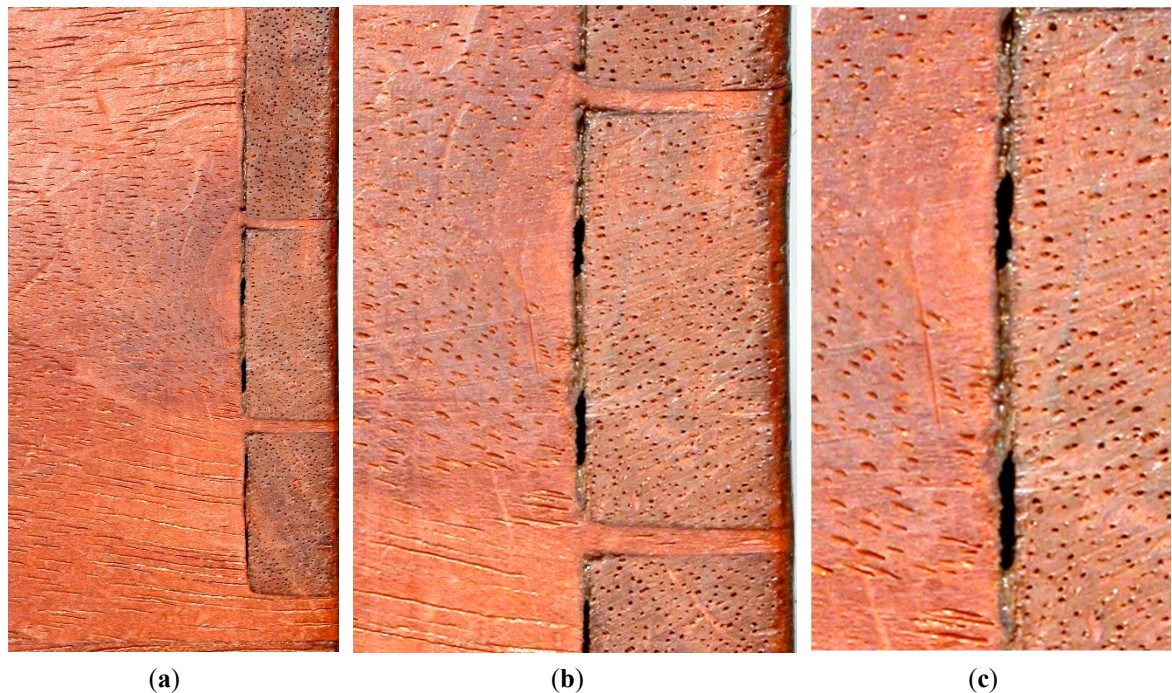

**(a)**       **(b)**       **(c)**

**Figure 12.** (**a**) Detail of the front right corner (about 1:2); (**b**) detail of the front right corner (about 1:1); (**c**) Detail of the front right corner (about 2:1).

8    Digital microscope RoHS, HD color CMOS, High Speed DSP, 24-bit DSP, Resolution 640×480×5×, digital zoom, magnification 1000× focus range 15–40 mm.

---

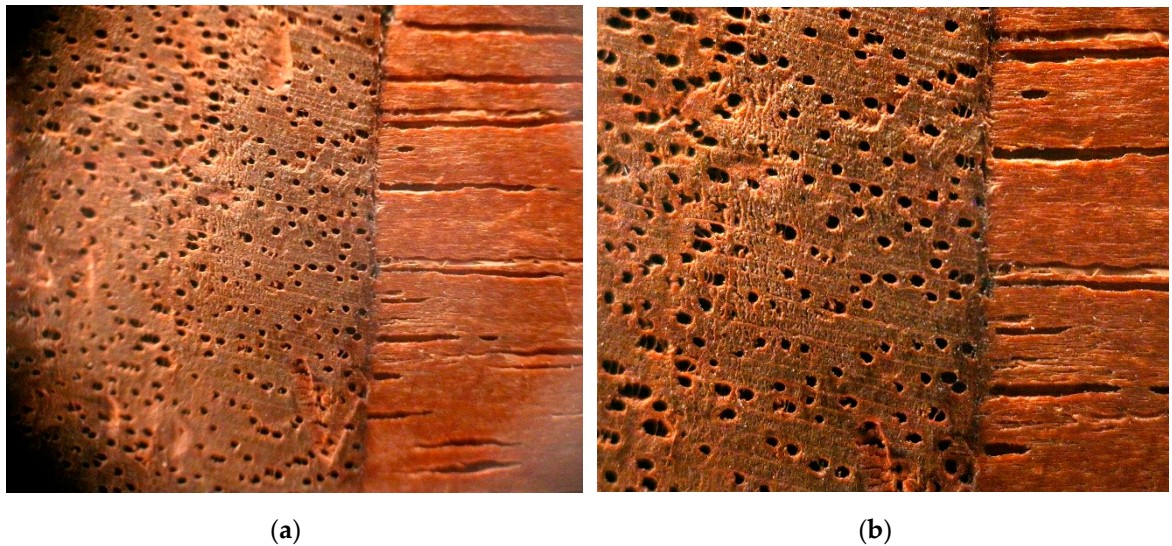

<div align="center">(<b>a</b>)                                   (<b>b</b>)</div>

**Figure 13.** (**a**) Detail of the front right corner (about 5×); (**b**) detail of the front right corner (about 10×).

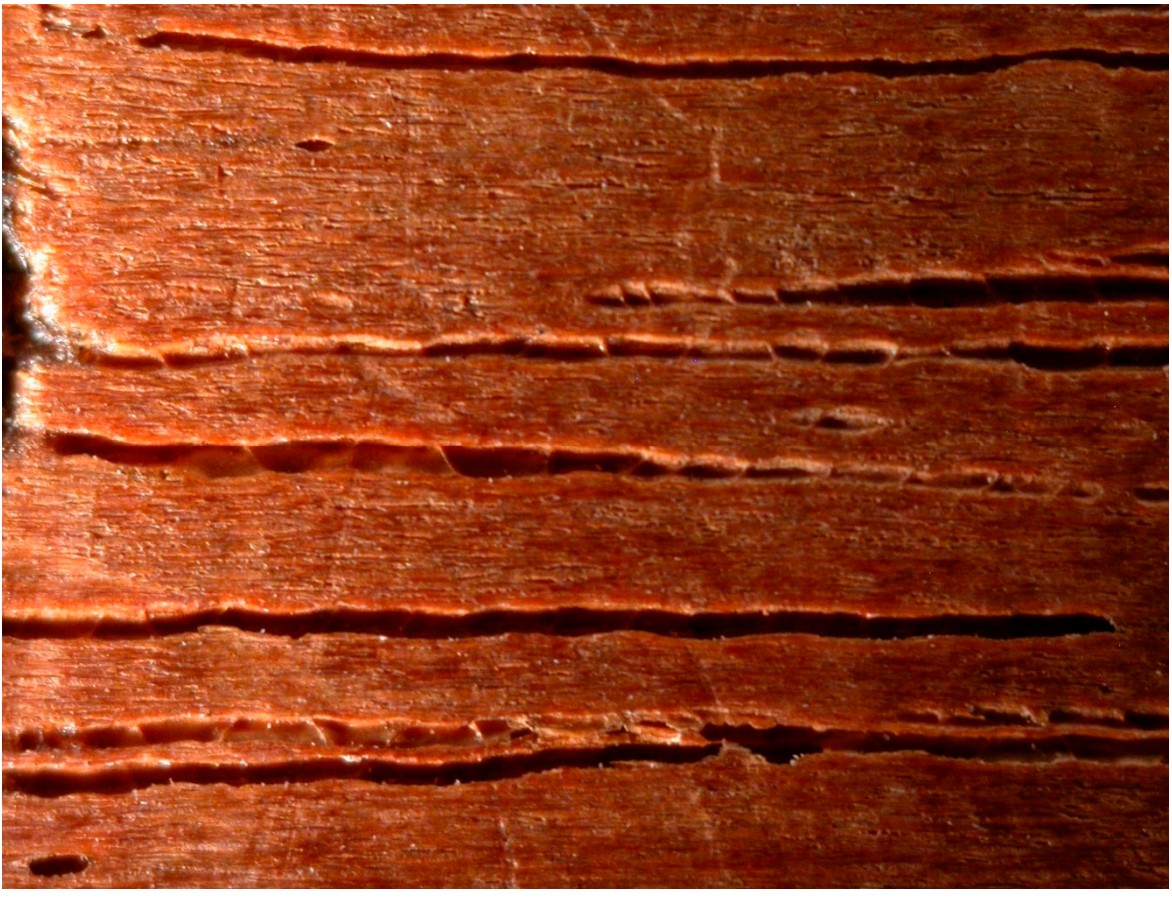

**Figure 14.** Enlarged detail (30×) of the longitudinal surface. Simple perforations of the transverse walls of the conductive tissues are clearly visible.

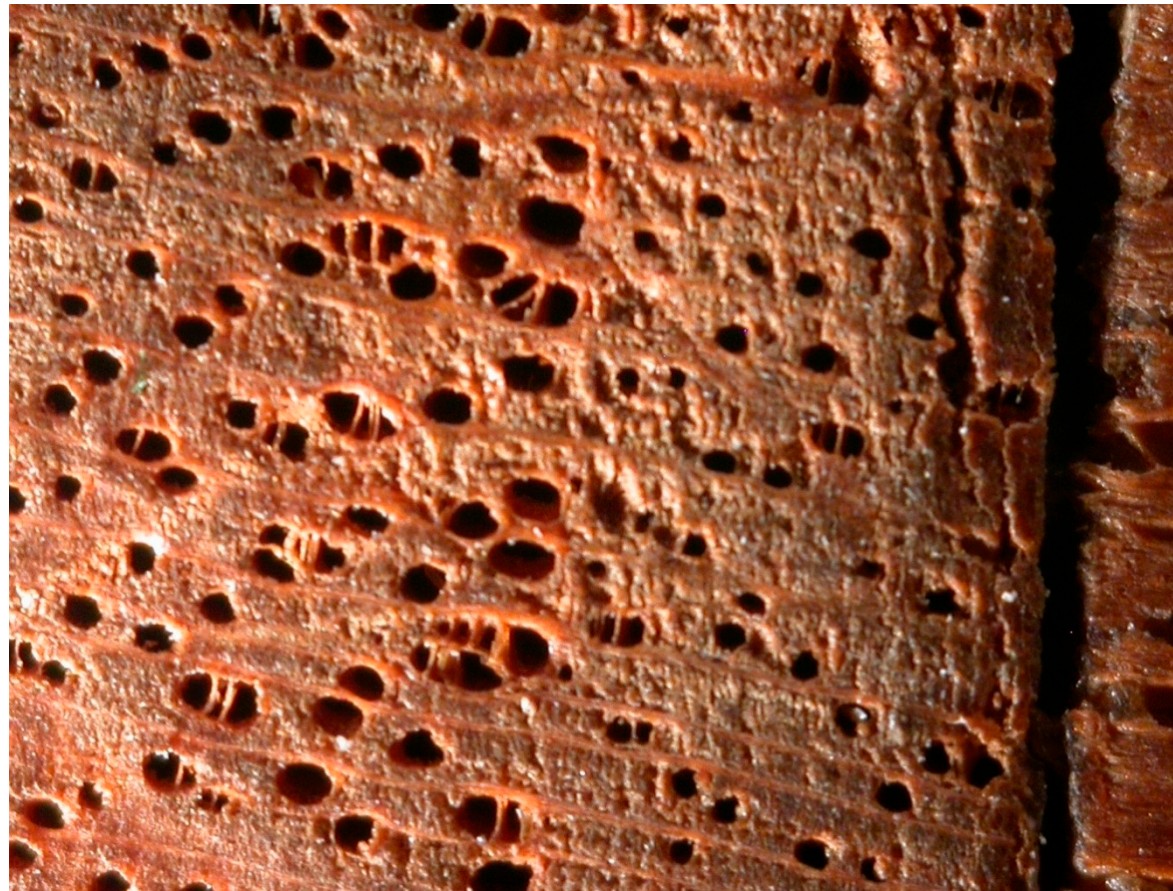

**Figure 15.** Enlarged detail (30×) of the transverse surface. The parenchyma rays and multiple combinations of conductive tissue are clearly visible.

### 3.3. Stylistic Elements

The only useful stylistic elements are constituted by the iron cartouche of the lock and by the walls' interlocking intervention with a dovetail system (Figure 16).

The iron cartouche of the lock is typically European (Figure 16a). It is more difficult provide chronological indications about the "dovetail" interlocking system because this system was used—especially in Europe—for many centuries (Figure 16b).

In any case, it is believed that none of these construction or stylistic elements might indicate a craftsmanship of the box different from European standards.

Stylistic research on traditional Chinese wooden box production of the Yuan dynasty were performed, but none of them looks even remotely similar to the studied box (Figure 17).

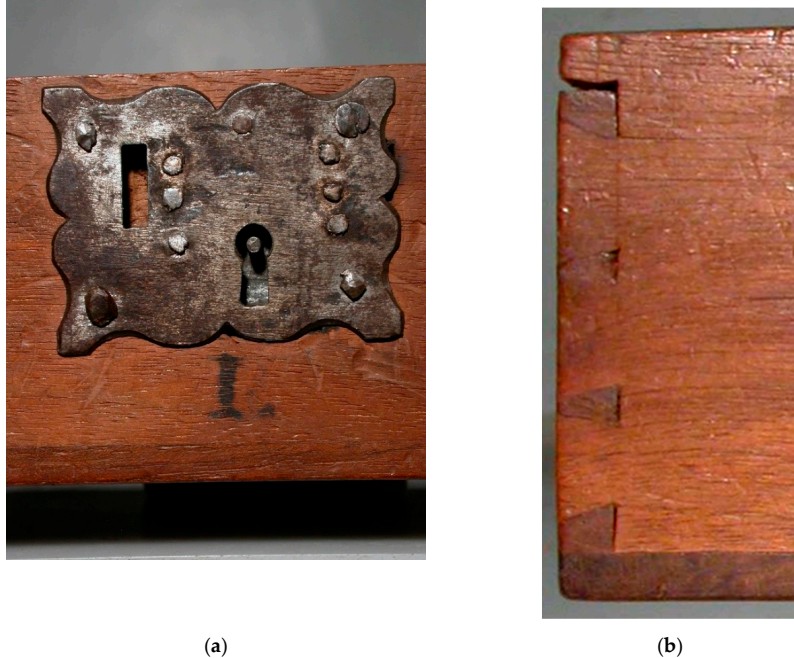

(**a**) (**b**)

**Figure 16.** (**a**) Detail of the lock's iron cartouche; (**b**) detail of the dovetail interlocking system.

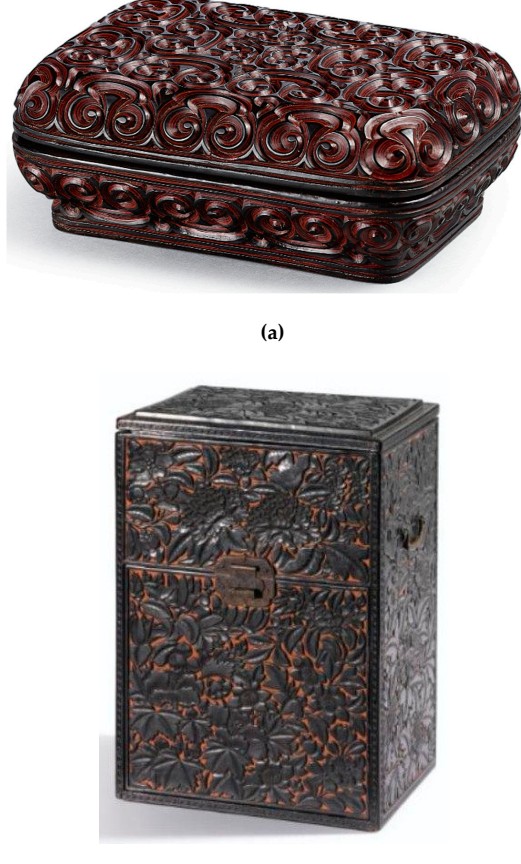

(a)

(b)

**Figure 17.** Examples of traditional Chinese wooden boxes of the Yuan dynasty. (**a**) A carved black and red Guri lacquer box[9]; (**b**) a carved black and cinnabar lacquer stationary box[10].

## 4. Conclusions

All the possible indicators used (e.g., construction characteristics, species of wood, finish, and style) suggest a European production of the box.

The iron cartouche of the lock is typical of European style, dated between the late seventeenth and early eighteenth century.

Elements such as wood species used and the iron cartouche—although uncertain—argue in favor of an Italian production.

These considerations suggest that the box could have been made to protect the Bible, which arrived in Italy already in poor condition after a long journey from China.

Alternatively, the box may have been built a few years after the arrival of the Bible in Florence, because it was in extremely unfavorable conditions of preservation, such as presenting an almost irreversible degradation (absolutely incompatible with the good conservation conditions of the box).

According to these plausible reasons, the box might have been built between the late seventeenth and early eighteenth centuries, in order to prevent possible page loss, waiting for a restoration that happened more than three hundred years later.

This work also demonstrated that a quick and inexpensive in situ survey, performed in inadequate conditions using only in-depth observation and simple tools without taking samples, is possible, and is particularly suited for potentially noteworthy relics that cannot accept any kind of material sacrifice.

**Funding:** This research was privately funded in 2012 by Fondazione per le Scienze Religiose "Giovanni XXIII", Bologna, Italy.

**Conflicts of Interest:** The author declares that there is no conflict of interest regarding the publication of this paper.

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
