# Peer review of "Studies on the Wooden Box Containing the “Marco Polo” Bible"

_heritage, doi:10.3390/heritage2010031_

Round 1

Reviewer 1 Report

The subject of the paper is an extraordinary artefact and I would suggest a more careful approach. Using IR spectroscopy and XRF (portable version) you can determine the chemical composition of the inks used, the composition of the wooden lake, and paper composition and you can identify if all of them are from the same period or are added later. An interdisciplinary team would be useful. The only conclusion of the works is the identification of the type of wood using microscopy. I think you need more than 5 references for a Q2 journal. The bibliography should be extinct with other studies identifying wood or pigments on artworks.

Author Response

Dear Reviewer,
thank you very much for your notes and suggestions.

Aim of the work has been also to demonstrate that a fast and cheaper survey, using only knowledge of wood and item typology, in a not perfect condition, is possible. So using an in deep observation and using simple tools as well as portable microscope without take samples. I didn't work on the Bible, I only worked on the wooden box. Other researchers used, in the restoration laboratory, instruments useful for determining chemical composition of paper and related ink. The result is that bible is for sure of the same period of Marco Polo.

Writings on the box are clearly datable (or already dated) without any kind of instrument.

About my work, as I told (but I will explain better it in the paper) I didn't want to touch that important box. And because my task was to work in rush and in a not adequate space (in a little room inside the Florence library) only to determine if the box was of the same period of the bible, I decided to avoid any sacrifice of material.

The owner didn't required a team for the box, team that actually worked on the bible judged most important than the box.

I agree with you about the needs of improving the references.

Reviewer 2 Report

The work regards the characterization of the origin of the wooden box employed to preserve an important manuscript probably belonged to Marco Polo of the Laurential Library of Florence. 

The study is poor in technical and hystorical contents. The statements and the conclusions are generally not supported by the comparison with references studies or by the results obtained by scientific investigations.

The description of the investigation on the wood species is the only part of the study dealt with sufficiently in depth.

The English language needs to be revised.

In particular:

-Line 10: the manuscript is indicated as "document". The difinition is imprecise since the manuscript is a library heritage.

-Line 17: "the East" is imprecise since the author wants to define if the wooden box has a chinese origin.

-Lines 30-33: from the historical point of view, the study of the "indirect sources" is of great relevance and can help in determining the hystorical events of a library heritage. I believe it would have been necessary to mention the archival documents regarding the manuscript.

-Fig. 1 and Fig. 2 are not useful for the study.

-Paragraph 3.1: the box is widely described. It would have been useful to discuss this hystorical information.

-Line 105, line 107, line 118, lines 121-122: the identification of the various types of glued is not supported by references or scientific investigations.

-Lines 123-124: the assertion is not supported by the comparison with references studies or particular specimens. The reference to Fig. 20 is not correct. Probably it is related to Fig. 16 where some chinese wooden boxes are reported without any hystorical or geographical contextualization. In the caption of Fig. 16 the source of the images is not indicated.

-Paragraph 3.2: the description of the investigation on the wood species is dealt with a just sufficient depth but the technical details of the microscope employed are not reported. 

-Lines 150-151: the wood species can indicate the provenance of the employed wood but no certainties on the place of the production.

-Paragraph 3.3: the stylistic elements investigation is not adequate. It lacks of some references supporting the conclusions of the author and of comparisons with typical specimens of the same period.

-Lines 180-181: the iron cartouche of the lock is indicated as typical of the 17th or 18th century while in the previous paragraph it is stated that it is difficult to provide chronological indications.

-Line 183: the Italian production, althought indicated as uncertain, is not supported by any hystorical or scientifc evidence. 

-Lines 189-190: the dating is not supported by relevant hystorical or scientific evidences.

Please control the references style.

The article cannot be acceptable in this form.

A major revision of the text and of the contents is recommended to be considered for pubblication.

Author Response

Dear Reviewer,
thank you very much for your notes, comments and suggestions.

I try to reply to your comments:

About the study that is poor in hystorical contents I agree but no historical documents about the box are available. Some historical information are strictly related (but not verificable) about the bible.

About the study that seems to be poor in technical contents, I tried to explain (but also I will try to explain better) that aim of the work has been also to demonstrate that a fast and cheaper survey, in a not perfect condition, is possible. So using an in deep observation and using simple tools as well as portable microscope without take samples. The macroscopic methodology I used is endorsed by technical standards as well as I referred to the reference number 4.

As I told I didn't want to touch that important box. And because my task was to work in rush and in a not adequate space (in a little room inside the Florence library) only to determine if the box was of the same period of the bible and if it was made in China, for that I decided to avoid any sacrifice of material.

About your comment that the statements and the conclusions are generally not supported by the comparison with references studies or by the results obtained by scientific investigations, I didn't find any research about ancient wooden boxes. I will try to find something again. About the scientific investigation of wooden object I referred to the UNI Standard, the only scientific tool available that admit macroscopic survey, mainly for wooden heritage items, to avoid sacrifices of material.
I will try to improve English language and references. I will also try to add the suggested informations and revisions.

Reviewer 3 Report

The paper focuses on the results of a research on a wooden box that holds an important historical document - a hand Bible handwritten in the 13th century, historically considered to have belonged to Marco Polo - aimed at evaluating its age and place of production. 

The topic is very relevant and requires uncommon specialist skills which are quite evident in the work 

The paper was drafted with methodological rigor and care; however, conclusions need to be slightly improved.

Author Response

Dear Reviewer,
thank you very much for your time and suggestions.

I will try to improve conclusions as well as the requests of the other reviewers.

Reviewer 4 Report

N/A

Author Response

Dear Reviewer,
thank you very much for your review.

I will try to improve English language, methods, results, conclusion and references.

I will add in the paper that:

Aim of the work has been also to demonstrate that a fast and cheaper survey, in a not perfect condition, is possible. So using an in deep observation and using simple tools without take samples.

I will try to explain better in the paper that I didn't want to touch that important box. And because my task was to work in rush and in a not adequate space (in a little room inside the Florence library) only to determine if the box was of the same period of the bible, for that I decided to avoid any sacrifice of material.

Round 2

Reviewer 1 Report

Dear author

I understand the intention to demonstrate "that a fast and cheaper survey, using only knowledge of wood and item typology, in a not perfect condition, is possible" but not always this approach leads to valid results. Please check the English : naked eye can be replaced with a better expression like free observation. Please check caption for figure Figure 12, magnification factor (5X an 10X) i think is inversed.

Author Response

Dear Reviewer,

I agree with you but the client request was a fast and in site survey.

I tried to improve something about the language and changed the term "naked eye" with "macroscopic view".

You right! During the editorial review the figures 12a and 12b have been inverted. I pointed out the need to place them correctly.

Thank you very much for your suggestions.

Sincerely

Francesco Augelli

Reviewer 4 Report

It is absolutely essential that an English-first-language copy-editor revise this manuscript. The topic is inherently interesting, and the manuscript is much improved. However, there is no substitute for employing a highly-skilled English-language proofreader in the preparation of the the final version.

Author Response

Dear Reviewer,

I tried to improve the English language.

Thank you for your availability.

Sincerely

Francesco Augelli